# Femtosecond electron beam probe of ultrafast electronics

Maximilian Mattes [1,2], Mikhail Volkov [1,2] ✉ & Peter Baum [1] ✉

The need for ever-faster information processing requires exceptionally small devices that operate at frequencies approaching the terahertz and petahertz regimes. For the diagnostics of such devices, researchers need a spatio-temporal tool that surpasses the device under test in speed and spatial resolution. Consequently, such a tool cannot be provided by electronics itself. Here we show how ultrafast electron beam probe with terahertz-compressed electron pulses can directly sense local electro-magnetic fields in electronic devices with femtosecond, micrometre and millivolt resolution under normal operation conditions. We analyse the dynamical response of a coplanar waveguide circuit and reveal the impulse response, signal reflections, attenuation and waveguide dispersion directly in the time domain. The demonstrated measurement bandwidth reaches 10 THz and the sensitivity to electric potentials is tens of millivolts or −20 dBm. Femtosecond time resolution and the capability to directly integrate our technique into existing electron-beam inspection devices in semiconductor industry makes our femtosecond electron beam probe a promising tool for research and development of next-generation electronics at unprecedented speed and size.

The demand for ultimate speed in modern information processing and data transfer has prompted extensive research on high-speed electronics at maximum frequencies and shortest switching times[1–15]. While most electronic devices currently operate at gigahertz frequencies, the leading edge of electronics verges into the terahertz[1,2] and even petahertz domain[3–8]. For example, terahertz technology and millimetre waves[9] are the basis of the forthcoming sixth-generation telecommunication standard (6 G)[1], and high-speed transistors[10–13], hybrid photonic platforms[14,15] or terahertz metadevices[2] begin to merge the electronic and optical domains. Numerical modelling above 100 GHz is difficult because the skin depth approaches the surface roughness[16], radiation losses strongly increase[17] and the concept of carrier mobility breaks down as the motion becomes ballistic[18]. Therefore, the experimental characterisation of local electromagnetic fields and their dynamics in such future devices so far remains an open challenge because the required bandwidth and time resolution can obviously not be provided by electronics itself. Characterising record-breaking devices requires a conceptionally different approach.

Modern research offers some creative diagnostic solutions to deal with the small size of devices or their high speed, but not at the same time. For example, sub-nanometre resolution can be provided by scanning tips[19] or electron beam probing (eBeam)[20], but the maximum bandwidth is limited to several gigahertz[21–24], far below the terahertz domain. Up-conversion of gigahertz signals with high-frequency transistors[11] and microwave analysis[25] provide frequency resolution of up to 1.1 THz[26], but these approaches cannot resolve the local fields inside the functional parts of a device. If optical techniques are combined with scanning probe tips for electro-optic[27] or photoconductive[28] sampling, scanning tunnelling microscopy[29] or electric force microscopy[30], a probe tip must be placed in close physical proximity to the device and inevitably disturbs the local fields. Femtosecond point-projection microscopy[31–33] is a promising development, but the specimen is not in a field-free region, and the low electron energies require free-standing materials.

Here, we show how an ultrafast electron microscope[34] with femtosecond electron beams[35–42] under the control of optical fields of laser

[1]Universität Konstanz, Universitätsstraße 10, 78464 Konstanz, Germany. [2]These authors contributed equally: Maximilian Mattes, Mikhail Volkov.
✉e-mail: mikhail.volkov@mbi-berlin.de; peter.baum@uni-konstanz.de

light[36] can probe the position-dependent and time-dependent local electric and magnetic fields of ultrafast electronic circuits while operating in the terahertz regime.

## Results

Figure 1 shows the concept and fundamentals of our experiment. A broadband impulse response of a device under test (DUT) is triggered by a laser-excited photoconductive switch[43,44]. The produced ultra-short voltage pulse propagates through the DUT and triggers its functionality in space and time. Femtosecond electron pulses for probing purposes are created by laser-driven two-photon photoemission[45] and subsequent acceleration to tens of keV[42]. If necessary for time resolution, these electron pulses are compressed in time with an optically generated terahertz field[36] to obtain sub-100-fs duration. Using magnetic lens systems, the femtosecond electron pulses are focused onto interesting parts of the DUT. There, the local electromagnetic fields in the DUT are probed directly in the time domain via deflection by time-frozen Lorentz forces[46,47]. The resulting beam deflections are observed on a screen and reveal the magnitude and vectorial direction of the local and time-frozen in-plane fields. Electric and magnetic components can be distinguished by their different dependencies on electron energy[46]. A variation of the delay between the DUT trigger and the femtosecond electron pulses by means of moveable mirrors and scanning the electron beam across the DUT then provide a stroboscopic movie of the local electromagnetic field vectors in space and time.

In the experiment, we investigate one of the key components for terahertz electronics, that is, a transmission line in the form of a coplanar waveguide on an insulating substrate, one of the most common ways for signal transport. Femtosecond electron pulses at a beam energy of 70 keV are produced by femtosecond laser photoemission[36,37] and subsequently compressed in time by terahertz radiation[36]. Magnetic lenses without temporal distortions[48] are used to focus and steer the beam onto the specimen for pump-probe experiments. Figure 2 shows an example of the measured beam deflection. A schematic of our device under test (DUT) is shown in Fig. 3a. Our circuit consists of (left to right) a bias pad, a photoconductive switch, a coplanar waveguide and a terminating rectangular pad with a 50-Ω shunt resistor. Undoped GaAs is used both as the insulating substrate of the coplanar waveguide and the active gap material in the photoconductive switch. The terahertz circuit is manufactured with

ultraviolet lithography and wet etching, following established designs[49]. The central conductor of the coplanar waveguide has a width of 30 μm and is separated by 20-μm gaps from two adjacent ground plates. The width of the photoconductive gap is 10 μm. Figure 3b shows an optical microscopic image of the fabricated structure, and Supplementary Fig. 1 shows additional results. The photoconductive switch is biased with a direct-current voltage source (Keithley 6517B, Tektronix) at a range of −15 V to +15 V. Femtosecond triggering is achieved by 80-fs long laser pulse with a wavelength of 515 nm and a pulse energy of ~100 nJ, focused to a 10-μm spot at the photoconductive gap. The electric fields of the generated pulses are mostly pointing along the y direction (blue and magenta arrows) and, therefore, deflect the electron beam, while the magnetic fields mostly point in the electron beam direction and, therefore provide no substantial effect.

At two locations, marked with magenta and blue ellipses in Fig. 3b, we thin the specimen for electron transparency by laser-drilling two 20-μm holes. The first probe region allows us to measure the voltage between the central transmission line and the ground. The second probe region enables measuring the voltage between the bias channel and the ground. The deflected beam electrons are detected on a fluorescent screen (TemCam-F416, TVIPS) and the beam positions are determined by a series of Gaussian fits (see Fig. 2). Time-zero of the apparatus is calibrated by all-optical streaking[36] or alternatively by measuring the relative time delay between two probing positions at given distances from the gap.

In the first experiment, we investigate the ultrafast dynamics of our DUT with uncompressed electron pulses at a duration of ~800 fs[36]. Figure 3c shows the measured ultrafast voltage dynamics in the two probe regions (blue and magenta circles in Fig. 3b). We can detect electric potentials as low as 30 mV or powers of −20 dBm. Figure 3d shows a zoom into the earliest part of the response. We mark the most distinct features of our data with time labels $t_1$–$t_8$. In the first probe region (magenta), the initial response of the circuit ($t_1$) is a rapid voltage change from 0 to 5 V with a measured 2-ps rise time (10−90%), eventually broadened by the 800-fs electron pulse duration. There is a delay of 3.6 ps with respect to the trigger of the photoconductive switch at 0 ps. After this impulsive rise, the voltage pulse decreases to 50% of its maximum value within ~45 ps ($t_2$). Two weak additional voltage peaks are observed at 80 ps and 87 ps ($t_3$), and a stronger peak emerges at 130 ps ($t_4$). In the second probe region (blue), the initial response ($t_5$) is also strong but this time negative impulsive voltage change from 0 V to −6 V. The fall time is 2 ps (10−90%). There is also a delay with respect to the trigger of the photoelectric switch, this time 0.9 ps ($t_5$). During the subsequent rise of the voltage back to the ground, there are again several additional voltage peaks, for example, at 25 ps ($t_6$), 65 ps ($t_7$) and 120 ps ($t_8$).

When the photoconductive switch is triggered by the laser pulses, the optical carrier−hole pairs make the gap conductive within femtosecond times, and the bias voltage is shortened to ground. Consequently, there emerges an electric field in the form of approximately a Heaviside function that subsequently propagates as a positive or negative electric field into the left and the right side of our coplanar waveguide. These dynamics and the time that is needed to reach the points of measurement (blue, magenta) explain the strong and ultrafast voltage changes in our two measurements at opposite polarity and different delays ($t_1$ and $t_5$). In the 1−4 ps that are needed to propagate from the source to the measurement, the laser-generated voltage pulse already attenuates and disperses to picosecond duration[47]. The measured rise time in Fig. 3d is ~2 ps and corresponds, if associated with a quarter wave period, to a frequency of ~0.12 THz, which is well below our sampling frequency of about 1/(800 fs) = 1.2 THz. The measured rise time is, therefore, not the time resolution of our experiment but rather the time-resolved dynamics of the waveform in

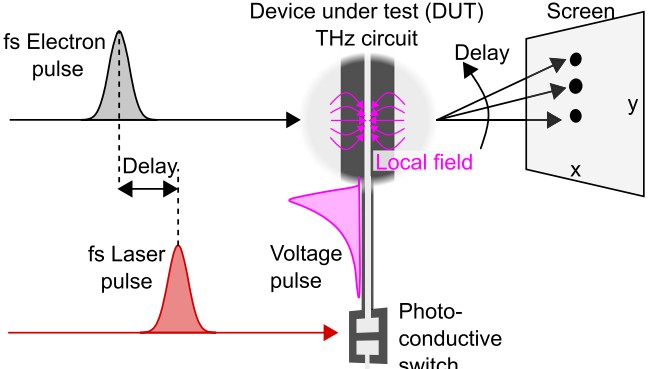

**Fig. 1 | Femtosecond electron beam probe of terahertz electronics.** A laser pump pulse with femtosecond duration (red) creates a terahertz voltage pulse (magenta) by closing a photoconductive switch. The terahertz voltage pulse then travels into the device under test (DUT) and triggers its operation. Femtosecond electron pulses (grey) probe the local electric and magnetic field vectors (magenta) by means of time-frozen Lorentz forces and electron beam deflections. Changing the delay between the activation of the switch and the probing electron pulses (dashed lines) provides a measurement with femtosecond time resolution and terahertz bandwidth.

our transmission line. Experiments with time-compressed electron pulses confirm this assessment; see below.

The delays of the two waveforms in Fig. 3d are caused by the propagation of the voltage pulse from the photoconductive switch to the probe region. The observed values of $t_1 = 3.6$ ps and $t_5 = 0.9$ ps, together with the physical distances of the two probe locations at $410\,\mu$m and $105\,\mu$m away from the photoconductive gap, reveal a group velocity of 0.38 c. This speed corresponds well to a simple estimate of pulse propagation speed in our coplanar waveguide, given by $c/\sqrt{(\varepsilon+1)/2} = 0.38$ c for a dielectric constant for GaAs of $\varepsilon = 13$ (ref. 50). The recovery time of the positive pulse propagating to the right (~45 ps) is longer than the recovery time of the negative pulse propagating to the left (~10 ps). We attribute this observation to the different effective capacities and ohmic resistances in the left and right parts of the coplanar waveguide. Additional contributions may also arise from a potentially asymmetric position of the laser beam profile on the gap as well as heating effects[51] or differences in the electron and hole mobilities via the photo-Dember effect.

The additional voltage peaks in both recovery signals ($t_3$, $t_4$, $t_6$–$t_8$) are time-domain signal reflections from geometry changes in our transmission line with partial impedance mismatch[51] (see arrows in Fig. 3a). The two minor voltage peaks in the right-propagating pulse (magenta) at $t_3 = 80$ ps, and 87 ps correspond to reflections at the interconnection to the shunt pad (see Fig. 3a), first into a bare Goubau line without adjacent ground lines and then into the macroscopic rectangular bond area. The stronger reflection observed at $t_4 = 130$ ps originates from the end of the shunt pad at a total distance of 11.06 mm. These two values reveal a slower average group velocity of 0.294 c as compared to 0.38 c in the transmission line because the shunt pad has a different size and distances to the ground, producing a lower cutoff frequency and large dispersion. The wirebond connections to the 50-Ω resistor are too thin to be relevant for our picosecond dynamics. On the left side of the coplanar waveguide (blue data), the voltage pulse in the negative voltage region at $t_6 = 30$ ps originates from impedance mismatch when the waveguide opens in a triangular way to the DC bias pad. The voltage pulse at $t_7 = 65$ ps is a reflection from the end of the bias pad. The latest measured voltage pulse in our scan range at $t_8 = 125$ ps is a double reflection within the DC bias pad (see Fig. 3a). All secondary voltage pulses at later times are increasingly broadened by dispersion and also increasingly attenuated by ohmic resistances and total propagation time.

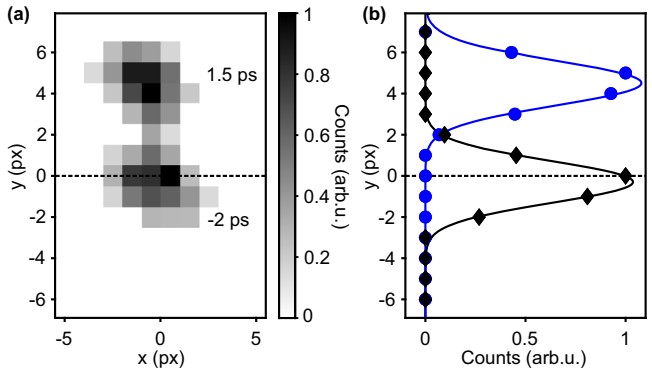

**Fig. 2 | Electron beam spots on the detector. a** Measured electron beam profiles at −2 ps and 1.5 ps after waveform generation. The dashed line is the equilibrium beam. **b** Gaussian fits (solid lines) reveal the position of the electron beam in $y$-direction (integrated over $x$). Black, reference beam; blue, beam at 1.5 ps.

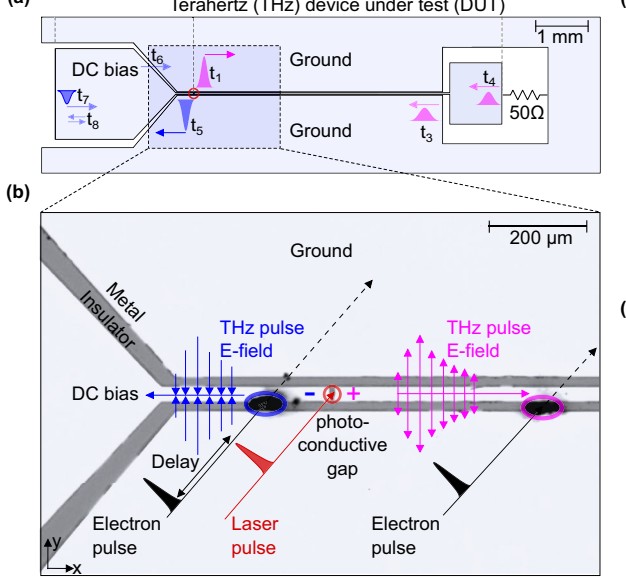

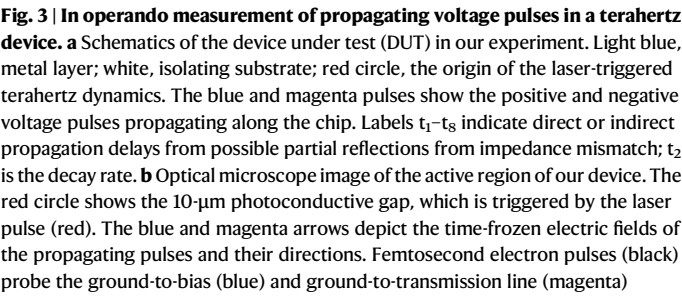

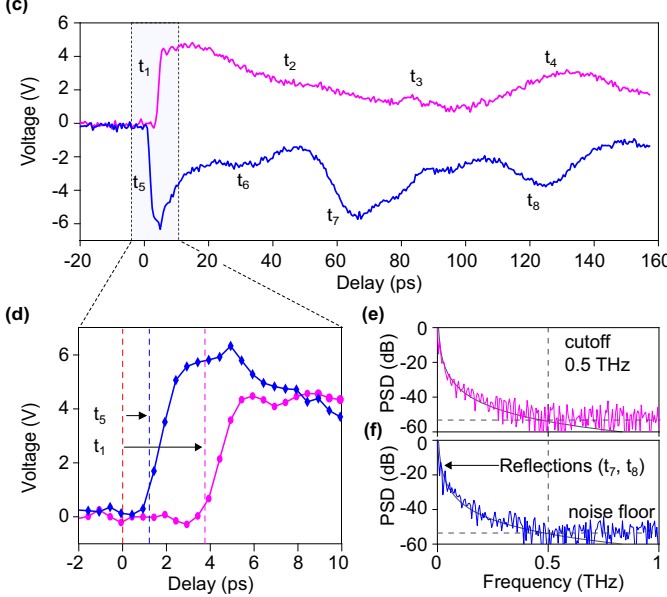

**Fig. 3 | In operando measurement of propagating voltage pulses in a terahertz device. a** Schematics of the device under test (DUT) in our experiment. Light blue, metal layer; white, isolating substrate; red circle, the origin of the laser-triggered terahertz dynamics. The blue and magenta pulses show the positive and negative voltage pulses propagating along the chip. Labels $t_1$–$t_8$ indicate direct or indirect propagation delays from possible partial reflections from impedance mismatch; $t_2$ is the decay rate. **b** Optical microscope image of the active region of our device. The red circle shows the 10-μm photoconductive gap, which is triggered by the laser pulse (red). The blue and magenta arrows depict the time-frozen electric fields of the propagating pulses and their directions. Femtosecond electron pulses (black) probe the ground-to-bias (blue) and ground-to-transmission line (magenta)

voltages at distances of −105 μm and 410 μm from the photoconductive gap, respectively. **c** Measured time-resolved voltage signals in the two probing regions. Labels $t_1$–$t_8$ indicate the rising slopes ($t_1$, $t_5$) and multiple reflections from the chip boundaries ($t_2$–$t_4$, $t_6$–$t_8$). **d** A zoom into the ultrafast slopes of both measured fields, plotted as absolute values. Measured delays of 0.9 ps and 3.6 ps are indicated with blue and magenta dashed lines. **e** Power spectral density (PSD) of the ground-to-transmission line voltage. Black line, theoretical model; dashed vertical lines, the cutoff frequency of our structure and noise floor. **f** Power spectral density (PSD) of the ground-to-bias voltage. Black line, theoretical model; dashed vertical lines, the cutoff frequency of our structure and noise floor.

**(a)**

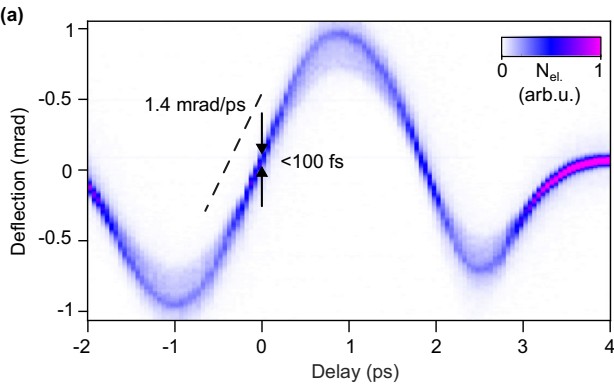

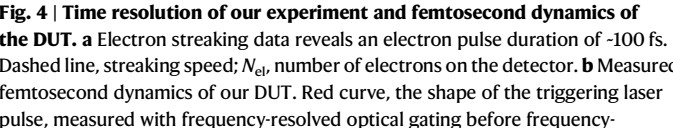

**(b)**

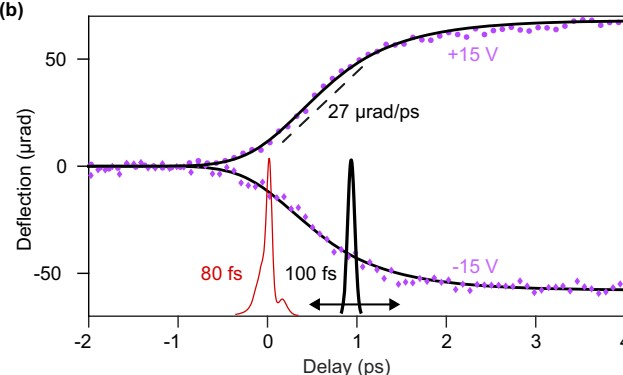

**Fig. 4 | Time resolution of our experiment and femtosecond dynamics of the DUT. a** Electron streaking data reveals an electron pulse duration of ~100 fs. Dashed line, streaking speed; $N_{el}$, number of electrons on the detector. **b** Measured femtosecond dynamics of our DUT. Red curve, the shape of the triggering laser pulse, measured with frequency-resolved optical gating before frequency-doubling. Black curve, 100-fs electron pulses. The measured device dynamics for positive (magenta dots) and negative bias (magenta diamonds) is fitted with a model (black lines) that accounts for the 100-fs instrumental function, a 720-fs exponential voltage buildup at the photoconductive gap and a 420-fs pulse elongation by waveguide dispersion.

Figure 3e, f shows the power spectral density (PSD) of the measured signals at the left and right side of the waveguide, evaluated by Fourier transformation from the time-domain results. We see broadband curves with cutoff frequencies of ~0.5 THz, where the response of our waveguide merges with the noise. This value agrees well with the cutoff frequencies observed in similar coplanar waveguides with electro-optical sampling[52]. The frequency spectrum of Fig. 3e, f is, therefore, the response of the device and is not limited by our time resolution. The multiple reflections from the left chip boundary ($t_5$, $t_7$, $t_8$) come in quasi-periodically and consequently show up in Fig. 3f as frequency peaks at ~16 GHz and its second harmonic at ~32 GHz. A pure step function, like initially created in the photoconductive gap, should have a $1/f$ spectrum, where $f$ is frequency, but our data fits better with a pink noise with $f^{-2.7}$ dependency (grey curves) because we measure the waveform after substantial dispersion from the femtosecond to the picosecond domain.

In a second experiment, we demonstrate the additional diagnostic capability that is gained by compressing the electron pulses from 800 fs to 100 fs with laser-generated terahertz fields[36]. The duration of the laser pulses (Fig. 1a, red) that trigger the photoconductive switch is still ~80 fs. In order to measure the electron pulse duration at the location of the DUT and thereby the time resolution of our ultrafast voltage probe experiment, we invoke an all-optical streak camera that is driven by the cycles of optically generated far-infrared light[36]. A bow-tie resonator is used to mediate the electron–photon interaction by providing energy-momentum conservation. Electrons are deflected sideways ($y$ direction in Fig. 3b) as a function of their arrival time with respect to one optical cycle of the reference wave. Consequently, long electron pulses produce a blurry streak while short ones produce a point and, therefore, sample the optical field cycle in an almost classical way[36]. Figure 4a shows the measured electron streaking trace[36] at the location of the DUT. We see a sinusoidal deflection of the beam electrons as a function of delay. The streaking speed around the zero crossing is 1.4 mrad/fs (dashed line). The width of the electron beam there, deconvoluted with the unstreaked electron beam profile[37], reveals an electron pulse duration of $100 \pm 10$ fs. The corresponding measurement bandwidth is ~10 THz.

Figure 4b shows the time-resolved dynamics of the electric field between the central transmission line and the ground at the right side of the gap (magenta ellipse in Fig. 3b). The data point separation is 80 fs and the Nyquist frequency is 6 THz. In this experiment, we apply a positive bias voltage of +15 V (magenta dots) and a negative bias of −15 V (magenta diamonds) to the bias pad. The time-resolved voltage dynamics in both cases (magenta dots and magenta diamonds) are

mirror images of each other, demonstrating the unidirectional response of our photoconductive switch. Like in the lower-resolution experiment of Fig. 3d, we can see an ultrafast rise or fall time of ~2 ps, but this time, the compressed electron pulses allow us to resolve the shape of this response and conclude on the broadening mechanisms in the time domain.

Supplementary Fig. 2 shows fitting attempts with a broadened Heaviside function or exponential growth. The residuals in both traces are substantial, and these models, therefore, do not capture the physics of our experiment. In order to fit the measured data in a proper way (black lines), we consider our 100-fs apparatus function in the time domain, an exponential rise time due to the finite capacity of our photoconductive gap and a dispersive pulse elongation from propagation in our transmission line. Consequently, we fit our data $U(t)$ by

$$U(t) = A e^{-\frac{t-t_0}{\tau_{RC}}} \mathscr{H}(t-t_0) \otimes e^{-\frac{t^2}{\tau_{disp}}} \otimes e^{-\frac{t^2}{\tau_{inst}}}, \qquad (1)$$

where $\mathscr{H}$ is the Heaviside function and $\otimes$ is a convolution in time. The fit parameters are the signal amplitude $A$, the time zero $t_0$, the time constant $\tau_{RC}$ of the photoconductive gap and the pulse broadening $\tau_{disp}$ due to propagation. The instrumental function of our electron beam probe $\tau_{inst}$ is fixed to 100 fs (Fig. 4a). Signal decay due to electron-hole recombination or damping in the transmission line can be ignored on this timescale because it takes tens of picoseconds, at least.

The best global fit for both bias cases (black lines) is obtained with an RC-limited rise time of $\tau_{RC} = 0.72$ ps and a dispersive broadening of $\tau_{disp} = 0.42$ ps. Fitting attempts with simpler models do not yield good results; see Supplementary Fig. 2. We assign the measured RC time constant to the electromagnetic response directly in the photoconductive gap, defined by the capacity of the structure and the effective mobility of the hot charge carriers after laser excitation, while the dispersion time reveals the effects of the first picoseconds of propagation within our transmission line. Although the gap, as well as the waveguide, can on their own support a femtosecond response, the accumulated effects add up to an effective rise time of ~2 ps at the location of the probe. These proof-of-principle results demonstrate the value of using femtosecond time resolution and oversampling of picosecond signals to extract and discern the underlying spatio-temporal broadening mechanisms for optimising future devices towards desired functionalities.

Given that electron pulses in electron microscopes can be made as short as attoseconds[39,40], what are the fundamental resolution limits of our approach? Since the local electromagnetic fields in a DUT always

have a finite spatial extent in the direction of the probing electron beam, the maximum detectable frequency is limited by the time of flight of the electrons through such a field. This aspect is known as the transit time effect in secondary-electron-based voltage contrast in a scanning electron microscope (SEM)[53]. However, the electron beam in our experiment travels at almost half of the speed of light and, therefore, needs a hundred to thousand times less time to pass through a material than the low-energy secondary electrons in a conventional SEM. Also, detection of the direct beam after transmission preserves the emittance and therefore allows one to detect positions and deflections at the same time[46]. For ultimate space-time resolution, the electron time-of-flight through the local fields with characteristic longitudinal dimensions $\Lambda$ in electron beam direction must be shorter than half a cycle period of the highest frequency $f_{max}$ to be determined in order to not smear out the temporal response. The electron beam must be smaller than the spatial inhomogeneities of the field. At these conditions, the electron gains via the Lorentz forces $F_L$ a sideways momentum change of $\Delta p_\perp \approx F_L \cdot 0.5 \cdot f_{max}^{-1}$. The factor 0.5 accounts for our desire to resolve half a cycle of $f_{max}$. Fields in features of lateral size $d$ in a DUT ($y$-axis in Fig. 3b) roughly penetrate into free space by $\Lambda \approx d$. For a voltage amplitude $\Delta V$ to be resolved, the electron deflection angle $\Delta\theta$ is given by $\Delta\theta = \frac{\Delta p_\perp}{p_0} \approx \frac{e\Delta V}{2p_0 df_{max}}$, where $p_0 \approx 276$ keV/c is the longitudinal electron momentum, and $e$ is the elementary charge. If we focus an electron beam of emittance $\epsilon$ into a spot size of diameter $d$ as a local probe, we obtain a minimum beam divergence angle of $\Delta\theta_{beam} \approx \epsilon/d$. This angle should not be much larger than the sample-induced deflection $\Delta\theta$ in order to provide a clear determination of beam shifts. Note that beams with $\Delta\theta_{beam} > \Delta\theta$ may still be resolved at sufficient ratio of signal to noise. Using $\Delta\theta_{beam} < \Delta\theta$ as a worst-case criterion, we obtain a maximum measurable frequency of $f_{max} \approx \frac{e\Delta V}{2\epsilon p_0}$. Interestingly, this speed limit does not depend on the size $d$ of the structures in a DUT because a smaller structure produces higher electric fields for deflection but also requires a more divergent electron beam to be resolved in space. In our proof-of-concept experiment, the beam emittance at the specimen is $\epsilon \approx 100$ pm·rad and a signal with 1-V features in a 20-µm gap (see Fig. 3b) can be resolved up to $f_{max} \approx 5$ THz or at a temporal resolution of ~200 fs (compare Fig. 4). However, modern pulsed electron beams from better sources than ours, for example from Schottky field emitters, have a beam emittance of ~5 pm rad[54,55] which allows one, assuming beam energy of 200 keV, to resolve DUT frequencies as high as 60 THz or features as short as 20 fs at 1 V signal strength.

Static electron beam testing is ubiquitous in the semiconductor industry and inevitable for research and development, quality control or failure analysis. The special electron microscopes in such eBeam testing facilities typically work with high-brightness field-emission sources similar to those in transmission electron microscopy. The only changes to convert such devices into ultrashort fs-eBeam diagnostics are a laser-triggered electron source and a DUT-triggering by a photoconductive switch. Alternatively, a terahertz function generator can be synchronized to a femtosecond laser at 10-fs precision[56].

## Discussion

These results and considerations show that an electron beam probe with ultrashort electron pulses (fs-eBeam) can resolve the electromagnetic response of working electronic devices in space and time. Conventional scanning electron beam probe[57,58], a cornerstone technology of modern electronics research[20], is therefore advanced to terahertz bandwidth and femtosecond time resolution. No physical contact or proximity of a sensing object is required to achieve these resolutions, and the method is, therefore, non-distortive and impedance-free. In order to provide the necessary electron transparency, substrates can be thinned or precision-cut with ion mills or a focused ion beam. For example, large membranes of almost arbitrary material can be prepared by a dimple grinder and ion beam[59].

Alternatively, a non-invasive probing of circuit surfaces could be realised by detecting the energy or angular distribution of backscattered high-energy electrons in scanning electron microscopy. In principle, the spatial resolution in our fs-eBeam technique is only limited by the ability to focus an electron beam to a tiny spot, typically 0.2 nm in electron microscopy. The highest frequency and lowest electric fields that we can detect are only limited by the brightness of the electron beam and the averaging time. In contrast to near-field probes[19,27,29,30], the electrons in our experiment do not influence the dynamics in the DUT, and measurements can, therefore, be made on running devices under normal operation conditions. In principle, electric and magnetic field components can be distinguished by repeating the experiment with different electron velocities[46] and electron beam accelerations or decelerations by longitudinal electric fields can be revealed by applying an electron energy filter[55].

Detecting the most basic quantity of electronics, electric and magnetic fields, in space and time at nanometre, femtosecond, and millivolt resolutions should help researchers discover hidden mechanisms and design advanced ultrafast electronics beyond the state-of-the-art.

## Methods

The terahertz circuit is produced on a 625-µm thick, undoped, double-side polished GaAs wafer (University Wafers) by ultraviolet lithography (Smartprint, Smartforce technologies) with a 1-µm thick photoresist layer (AZ MIR 701, Merck). The design of the structure is based on a 50 Ω impedance line, and the dimensions are chosen to minimise losses[27]. After developing (MIF, Merck), the terahertz structure is deposited by thermal evaporation of 300 nm of gold. A 5-nm thick layer of chromium is used for adhesion. Lift-off is done by dissolving the sacrificial resist layer in acetone. The holes for the electrons are then drilled by using 1030-nm femtosecond laser pulses (Pharos, Light Conversion) focused to 10 µm diameter with an $f = 500$ mm lens. Drilling takes about 3 s at about 3 W average power. The chip is then contacted with the bias source with silver paste, and a 50 Ω resistor between the ground and signal is contacted with wire bonds. Supplementary Fig. 1 shows optical and scanning electron microscopy images of our circuitry.

Femtosecond electron pulses are produced by two-photon photoemission from a back-illuminated gold cathode at 70 keV[36,37] by frequency-doubled femtosecond laser pulses of 270 fs pulse duration (Pharos, Light Conversion). The centre wavelength for photoemission is 515 nm, and the repetition rate is 50 kHz. The average number of electrons per pulse is kept below ten to avoid dispersion by space-charge effects[42]. The emittance of the electron beam at the source is $\epsilon \approx 2$ nm rad[60] but later improved to ~100 pm rad by using a 20-µm aperture in a 100-µm beam. The electron pulses are compressed to ~100-fs duration by velocity-matched transmission through a metal membrane[37] under illumination with terahertz radiation that is produced by Cherenkov radiation in a LiNbO$_3$ slab[61]. All magnetic lenses for beam control are aligned for minimum temporal distortions[48]. The detector at 1.4 m distance from the specimen is a phosphor screen with a charge-coupled-device camera (TemCam-F416, TVIPS) with a pixel size of 15.6 µm. The electron pulse duration at the specimen and time-zero are characterised by all-optical streaking at a bow-tie resonator[36]. The pump laser pulses for triggering the DUT are compressed from 270 fs to 80 fs by cascaded $\chi^{(2)}$ interactions[62] and then frequency-doubled to 515 nm wavelength for efficient carrier-hole pair creation in the photoconductive switch. Measured electron beam intensities (integrated luminescence of the phosphor screen) in Fig. 2 are normalised to one. The relation between measured electron beam deflections and absolute voltages is calibrated with the help of the bias voltage; we find a linear response for the beam deflection with bias voltage in the entire range from −15 V to +15 V. The fitting error is computed as the residuals quadrature and minimised with a

derivative-free Nelder–Mead simplex algorithm. By varying the initial guess parameters, we ensure consistent convergence of the fit and emergence of a global minimum (see Supplementary Fig. 2).

## Data availability

The datasets used for the figures in this study are available from the corresponding authors upon request.

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

## Acknowledgements
This project has received funding from the Deutsche Forschungsgemeinschaft (DFG) via Sonderforschungsbereich SFB 1432, from the European Union's Horizon 2020 research and innovation programme under the Marie Skłodowska-Curie grant agreement No. 896148-STMICRO and from University Konstanz' Young Scholar Fund. We thank Stefan Eggert and Matthias Hagner for their help with specimen preparation.

## Author contributions
P.B., M.M. and M.V. designed the experiment. M.M. and M.V. performed the measurements and analysed the data. All authors wrote the paper.

## Funding

## Competing interests
The authors declare no competing interests.
