## [Peer Review File · Nature Communications]

Femtosecond electron beam probe of ultrafast electronicsREVIEWER COMMENTS

Reviewer #1 (Remarks to the Author):

The manuscript by Mattes et al. describes an interesting new in operando pump-probe possibility with which electronic processes can be probed with femtosecond time resolution with the help of an external electron pulse in ultrafast circuitry. In addition to demonstrating such a novel apparatus, the authors present proof-of-principle experiments in a coplanar waveguide and measure valuable data on impulse response, reflections, attenuation and dispersion. Because of the novel methodology presented and its expected impact on high-speed electronics, I definitely find this work worthy of publication in Nature Communications provided that the authors revise their manuscript according to the following comments and open questions in the manuscript.

1. In the chapter „Principle of...” more details should be given on the properties of the produced ultrashort voltage pulse so that the readers can assess the context of this work more accurately.
2. The authors state that the 2 probe positions with 20-micron diameter were fabricated by laser drilling. What was the reason that these holes were not prefabricated already upon the original UV lithography process for sample preparation? This would have resulted in better controlled structures and probe results.
3. The claim that I do not fully understand is the following. The authors claim to measure a 2 ps rise time of a voltage pulse with the help of a 800-fs electron bunch. Since the difference between the measured dynamics and the probe pulse duration is small, deconvolution should obviously be performed. However, the manuscript does not describe any details of such a procedure. This should be discussed upon revision.
4. Fig. 2(a) does not indicate t_2 , so this should be given there.
5. When the authors repeat their measurements with compressed 100-fs electron bunches, they again claim a 2 ps rise/fall time of their circuitry. In this case, deconvolution effects should be negligible, however, this then again prompts for more details to be given related to point #3 above.
6. The authors claim rise times and dispersive broadenings based on a 4-parameter fit saying that the acquired values were results of a “best global fit”. In a 4-dimensional parameter space this might sound slightly far-fetched, so I am wondering what are the other local minima that the search algorithm found in this parameter space and what are the corresponding time constants in these cases. Details of the fitting should also be explicitly stated in the Methods section.
7. I particularly appreciate the analysis in the “Resolution and speed limits” section which mainly deals with temporal resolution limits. The authors should also discuss any potential spatiotemporal coupling effects that might influence spatial and temporal resolution limits at the same time.

Reviewer #2 (Remarks to the Author):

The manuscript presented by the authors touches an important and timely topic. The solution proposed is original and relevant for the community. However, I have the feeling that it is difficult for me to trust what is written in the manuscript due to an important lack of statistics. I would like to see all the data behind this article to make sure the system works as claimed. Figure 2a shows very few data and the only one that seems to contain statistical validity is Figure 3b, but we only see the points extracted from the measurement but not the measurement itself. I think this should be included in the supplementary information.

Another thing that bothers me is that there is not enough experimental information. The type of circuit fabricated, and all the steps followed for its fabrication must be described in detail in the supplementary information, including schematic and optical microscope images. In general, I see too many drawings and too few real images of what has been fabricated. Giving more data of how the electron emission is done is also necessary for the readers to replicate the experiments (please provide images when suitable).

If the authors add the information required, I will have no problem on recommending its publication in Nature Communications. But as it is right now, many readers could get confused and it would be impossible to reproduce the work.

Reviewer #3 (Remarks to the Author):

The authors use a short (~ 80 -fs) optical pulse to trigger a voltage pulse that propagates along a microwave transmission line. The electric field that is associated with the voltage pulse is probed at two different positions along the transmission line (separated by a few $100\ \mu\text{m}$) using a pulsed electron beam. The local electric field deflects the electron pulse and the electron beam deflection is measured at the two positions as a function of the time delay between the optical driving pulse and the probing electron pulse. A time delay of ~ 3 ps between is observed between the onsets of the deflection signals at the two positions along transmission line. This delay reflects a finite propagation speed of the voltage pulse of $\sim 0.3\ c_0$, as expected from classical waveguide theory. I acknowledge the current interest in probing the spatiotemporal dynamics of local optical fields in nano- and microstructured antennas using time-resolved electron microscopy. Nevertheless, I do not think that the manuscript meets the criteria for publication in Nature Communications for the following reasons:

- (a) It is not obvious that the reported experiments provide new insight into the propagation of voltage pulses along transmission lines. The reported propagation speeds are expected based on well-established knowledge about such transmission lines.
- (b) The ballistic propagation of local electric fields in nanoantennas has recently been probed using point-projection electron microscopy with substantially higher spatial and temporal resolution (G. Hergert et al, Light (2017), A. Wöste et al., Nano Lett. (2023)). Point-projection microscopy has also been used to study charge-separation in nanowires (M. Müller et al., Nature Comm. (2014). Earlier studies of related effects are cited in these papers. These results are not cited in the present work.
- (c) Since the experiments are performed in a transmission geometry, they require rather thin samples that are sufficiently transparent for tens of keV electron pulses. This limits the applicability of the method.
- (d) The experimental data probe the voltage pulse propagation only at two specific points along the transmission line and thus necessarily give rather limited insight into the spatiotemporal propagation dynamics.

I was somewhat confused by the terminology chosen by the authors. They write in the abstract (and also later in the text) that they "sense ... electro-magnetic fields ... with ... millivolt resolution". Common units of electric field are V/m. Therefore it is not obvious what is meant with "millivolt resolution".

It is not clear whether the experiment probes electric, magnetic or (as claimed in the abstract) electromagnetic fields.

The spatial characteristics of the voltage pulse propagating along the transmission line are not discussed.

The section "resolution and speed limits" is rather lengthy and not fully substantiated by the experimental results shown in the main part of the manuscript.

For these reasons, I think that the manuscript is better suited for a more specialized journal.

Referee #1 (Remarks to the Author):

The manuscript by Mattes et al. describes an interesting new in operando pump-probe possibility with which electronic processes can be probed with femtosecond time resolution with the help of an external electron pulse in ultrafast circuitry. In addition to demonstrating such a novel apparatus, the authors present proof-of-principle experiments in a coplanar waveguide and measure valuable data on impulse response, reflections, attenuation and dispersion. Because of the novel methodology presented and its expected impact on high-speed electronics, I definitely find this work worthy of publication in Nature Communications provided that the authors revise their manuscript according to the following comments and open questions in the manuscript.

Thank you for your recommendation.

1. In the chapter „Principle of...” more details should be given on the properties of the produced ultrashort voltage pulse so that the readers can assess the context of this work more accurately.

We agree and give now more details about the specimen and the field directions in the gap. We also discuss now how the magnetic components can be revealed by repeating the experiment at different electron velocities.

2. The authors state that the 2 probe positions with 20-micron diameter were fabricated by laser drilling. What was the reason that these holes were not prefabricated already upon the original UV lithography process for sample preparation? This would have resulted in better controlled structures and probe results.

Drilling produces some small amount of dirt and we found it easier to first produce a clean lithography and then drill instead of doing lithography on partially contaminated surfaces. We now describe the specimen fabrication procedure in much more details in the methods section.

3. The claim that I do not fully understand is the following. The authors claim to measure a 2 ps rise time of a voltage pulse with the help of a 800-fs electron bunch. Since the difference between the measured dynamics and the probe pulse duration is small, deconvolution should obviously be performed. However, the manuscript does not describe any details of such a procedure. This should be discussed upon revision.

This is correct and we now state that the measured 2-ps rise time is broadened by the 800-fs electron pulse duration in this experiment, and it is approximate. A deconvolution is not applied because we later measure the same dynamics at much higher time resolution.

4. Fig. 2(a) does not indicate t_2 , so this should be given there.

In the revised paper, we now mention t_2 in the figure caption.

5. When the authors repeat their measurements with compressed 100-fs electron bunches, they again claim a 2 ps rise/fall time of their circuitry. In this case, deconvolution effects should be negligible, however, this then again prompts for more details to be given related to point #3 above.

As we can see from the fits, the trace in the high-resolution experiment is non-exponential, and a simple rise time is not enough to describe it. We mention this now more clearly in the revised text. We also

now explain the fitting procedure in more details in the methods section and also show a new extended data figure 2 with a map of fit residuals.

6. The authors claim rise times and dispersive broadenings based on a 4-parameter fit saying that the acquired values were results of a “best global fit”. In a 4-dimensional parameter space this might sound slightly far-fetched, so I am wondering what are the other local minima that the search algorithm found in this parameter space and what are the corresponding time constants in these cases. Details of the fitting should also be explicitly stated in the Methods section.

The fitting error is computed as the residuals quadrature and minimized with a derivative-free Nelder-Mead simplex algorithm. By varying the initial guess parameters, we ensure consistent convergence of the fit. We now report in the revised manuscript a detailed description of this procedure, and a new extended data figure 2 shows maps of fitting residuals that reveal a single global minimum on all cases.

7. I particularly appreciate the analysis in the “Resolution and speed limits” section which mainly deals with temporal resolution limits. The authors should also discuss any potential spatiotemporal coupling effects that might influence spatial and temporal resolution limits at the same time.

Thank you for appreciating this section which we amended even further according to reviewer #3. We also now discuss potential spatiotemporal coupling effects.

Reviewer #2 (Remarks to the Author):

The manuscript presented by the authors touches an important and timely topic. The solution proposed is original and relevant for the community.

Thank you for your recommendation.

However, I have the feeling that it is difficult for me to trust what is written in the manuscript due to an important lack of statistics. I would like to see all the data behind this article to make sure the system works as claimed. Figure 2a shows very few data and the only one that seems to contain statistical validity is Figure 3b, but we only see the points extracted from the measurement but not the measurement itself. I think this should be included in the supplementary information.

The new Fig. 2 now shows the raw data of our experiment, that is, the measured electron beam spots, their profiles and the Gaussian fits that are used to extract the deflection angles that are later plotted in Figures 3c, 3d, 3e-f and 4b. We also describe this procedure now more clearly in the main text.

Another thing that bothers me is that there is not enough experimental information. The type of circuit fabricated, and all the steps followed for its fabrication must be described in detail in the supplementary information, including schematic and optical microscope images. In general, I see too many drawings and too few real images of what has been fabricated. Giving more data of how the electron emission is done is also necessary for the readers to replicate the experiments (please provide images when suitable).

The revised paper now explains the sample preparation in more detail, and a new extended data figure 1 shows two additional real images, one with optical microscopy and one with a scanning electron microscope. We also explain the electron emission process now in more details in the methods section.

If the authors add the information required, I will have no problem on recommending its publication in Nature Communications. But as it is right now, may readers could get confused and it would be impossible to reproduce the work.

We hope that the additional figures and supplementary information make our paper now acceptable for publication.

Reviewer #3 (Remarks to the Author):

The authors use a short (~ 80 -fs) optical pulse to trigger a voltage pulse that propagates along a microwave transmission line. The electric field that is associated with the voltage pulse is probed at two different positions along the transmission line (separated by a few $100\ \mu\text{m}$) using a pulsed electron beam. The local electric field deflects the electron pulse and the electron beam deflection is measured at the two positions as a function of the time delay between the optical driving pulse and the probing electron pulse. A time delay of $\sim 3\text{ps}$ between is observed between the onsets of the deflection signals at the two positions along transmission line. This delay reflects a finite propagation speed of the voltage pulse of $\sim 0.3\ c_0$, as expected from classical waveguide theory. I acknowledge the current interest in probing the spatiotemporal dynamics of local optical fields in nano- and microstructured antennas using time-resolved electron microscopy.

Thank you for appreciating the relevance of our work.

Nevertheless, I do not think that the manuscript meets the criteria for publication in Nature Communications for the following reasons:

(a) It is not obvious that the reported experiments provide new insight into the propagation of voltage pulses along transmission lines. The reported propagation speeds are expected based on well-established knowledge about such transmission lines.

The primary topic of our paper is the reported new concept for femtosecond time resolution in electron beam probe by transmission electron microscopy. We then continue to demonstrate these ideas with proof-of-principle experiments that demonstrate the capabilities and limits of our ideas. In the investigated transmission line, some measurement results are indeed expected but some are not, for example the 0.72 -ps response time of the photoconductive switch. In the revised paper, we now state more clearly that we do a proof-of-principle experiment to demonstrate a novel type of experiment.

(b) The ballistic propagation of local electric fields in nanoantennas has recently been probed using point-projection electron microscopy with substantially higher spatial and temporal resolution (G. Hergert et al, Light (2017), A. Wöste et al., Nano Lett. (2023)). Point-projection microscopy has also been used to study charge-separation in nanowires (M. Müller et al., Nature Comm. (2014). Earlier studies of related effects are cited in these papers. These results are not cited in the present work.

These papers are indeed relevant and we now cite them in the revised version of our manuscript.

(c) Since the experiments are performed in a transmission geometry, they require rather thin samples that are sufficiently transparent for tens of keV electron pulses. This limits the applicability of the method.

Yes, and we discuss this fact now more clearly in the revised version of our manuscript. We argue that modern TEM preparation techniques are mature enough to overcome this inconvenience. For example, large membranes of almost arbitrary material can be prepared by a dimple grinder and ion beam [doi.org/10.1063/1.5006522]; we now refer to this procedure.

(d) The experimental data probe the voltage pulse propagation only at two specific points along the transmission line and thus necessarily give rather limited insight into the spatiotemporal propagation dynamics.

The primary topic of our paper is the reported new concept for femtosecond time resolution in electron beam probe by transmission electron microscopy. In the revised paper, we now state more clearly that we do a proof-of-principle experiment to demonstrate the general feasibility of our novel type of experiment. With a larger thinned-out region, an entire trace can be obtained.

I was somewhat confused by the terminology chosen by the authors. They write in the abstract (and also later in the text) that they “sense ... electro-magnetic fields ... with ... millivolt resolution”. Common units of electric field are V/m. Therefore it is not obvious what is meant with “millivolt resolution”.

Thank you for pointing out this mistake; we now give the accuracy in terms of potentials in mV.

It is not clear whether the experiment probes electric, magnetic or (as claimed in the abstract) electromagnetic fields.

We measure the sideways components of the electric and the magnetic fields. They can be distinguished by repeating the experiment at two or three different electron beam energies; see Ryabov et al. Science 2016. In the revised paper, we discuss now more clearly what components of the electromagnetic field is actually revealed. Also, we give now a clearer outlook on how eventually the so far missing field components can in principle be determined by different electron velocities and electron energy analysis.

The spatial characteristics of the voltage pulse propagating along the transmission line are not discussed.

We now explain in more detail the direction of the electric and magnetic fields in our geometry and give more emphasis to the blue and magenta arrows of Fig. 3b.

The section “resolution and speed limits” is rather lengthy and not fully substantiated by the experimental results shown in the main part of the manuscript.

We aim here for providing the general reader with an assessment of the capabilities and limits of our new method for future application regimes. We find these thoughts important to understand the physics behind our ideas; compare reviewer #1 point 7.

For these reasons, I think that the manuscript is better suited for a more specialized journal.

We hope that the above explanations, amendments and revisions make our paper now acceptable for publication.

REVIEWER COMMENTS

Reviewer #1 (Remarks to the Author):

The authors have answered the comments of the reviewers in a satisfactory way. The manuscript is suitable for publication in Nature Communications.

Reviewer #2 (Remarks to the Author):

The revision made by the authors is quite borderline. The explanations added are very limited, the information provided does not clarify much more the questions that I raised in my previous report. For example, the new Figure 2 added has an axis in arbitrary units, which is very ambiguous. The authors should be much more generous on the explanations given, avoid references in the methods section (instead, describe the information cited), give the names of all the machines used (how the holes are drilled remains unclear), and describe everything what the readers need to do to reproduce the results. The rebuttal letter must contain the sentences included in the text that answer each response, so that it is crystal clear which is the effort made by the authors to provide the information requested.

Reviewer #3 (Remarks to the Author):

The authors have responded to my questions and comments. I now support publishing this work in Nature Communications.

Referee #1 (Remarks to the Author):

The authors have answered the comments of the reviewers in a satisfactory way. The manuscript is suitable for publication in Nature Communications.

Reviewer #2 (Remarks to the Author):

The revision made by the authors is quite borderline. The explanations added are very limited, the information provided does not clarify much more the questions that I raised in my previous report. For example, the new Figure 2 added has an axis in arbitrary units, which is very ambiguous.

The intensity scale in Fig. 2a and the horizontal axis in Fig. 2b are measured electron beam profiles on the screen, normalized to a peak intensity of one. The quantity behind it is the intensity of the green phosphorescence light from the scintillator screen multiplied by the sensitivity of the CCD electronics. These conversion factors are not available nor important, and they do not affect the evaluated quantity, position of the beam, in any way. We therefore argue that arb. units are appropriate and useful here. The procedure is now explained in the methods section.

The authors should be much more generous on the explanations given, avoid references in the methods section (instead, describe the information cited), give the names of all the machines used (how the holes are drilled remains unclear), and describe everything what the readers need to do to reproduce the results.

References are given wherever our setup or procedures are not new. We now give the name of all machines in the methods section. The hole drilling laser is a Pharos from Light Conversion with an $f = 500$ mm lens. We cannot see any remaining missing information that would prevent a team of fellow researchers in material science to reproduce our results.

The rebuttal letter must contain the sentences included in the text that answer each response, so that it is crystal clear which is the effort made by the authors to provide the information requested.

Attached is the revised methods section with all changes and additions marked in red.

Reviewer #3 (Remarks to the Author):

The authors have responded to my questions and comments. I now support publishing this work in Nature Communications.